# Priolog: Mining Important Logs via Temporal Analysis and Prioritization

**Byungchul Tak [1],\* , Seorin Park [1] and Prabhakar Kudva [2]**

[1]   Department of Computer Science and Engineering, Kyungpook National University, 80 Daehakro, Bukgu, Daegu 41566, Korea; befly098@knu.ac.kr
[2]   IBM TJ Watson Research Center, 1101 Kitchawan Rd, Yorktown Heights, NY 10598, USA; kudva@us.ibm.com
\*   Correspondence: bctak@knu.ac.kr; Tel.: +82-53-950-7565

**Abstract:** Log analytics are a critical part of the operational management in today's IT services. However, the growing software complexity and volume of logs make it increasingly challenging to mine useful insights from logs for problem diagnosis. In this paper, we propose a novel technique, Priolog, that can narrow down the volume of logs into a small set of important and most relevant logs. Priolog uses a combination of log template temporal analysis, log template frequency analysis, and word frequency analysis, which complement each other to generate an accurately ranked list of important logs. We have implemented this technique and applied to the problem diagnosis task of the popular OpenStack platform. Our evaluation indicates that Priolog can effectively find the important logs that hold direct hints to the failure cause in several scenarios. We demonstrate the concepts, design, and evaluation results using actual logs.

**Keywords:** log analysis; problem diagnosis; temporal correlation; log template; hierarchical clustering

---

## 1. Introduction

As the scale of modern IT services grows with increasing diversity of component architectures and behavioral patterns, the mission of seamless operation and efficient management becomes more challenging. A primary technique of achieving such goals is through log analytics [1–6]. The logs, generated from the system software, middleware, as well as applications, are the continuous stream of textual information that encode internal states of running applications. They are typically found in `/var/log` directory as text files in Linux, although locations are configurable. The capability to analyze logs is indispensable for enterprise-grade IT service management today. The log messages can be used as direct hints to the status or problems of services, or they can be viewed as a generic time-series data to which to apply time-series analysis to learn interesting temporal patterns. Although the ability to analyze the logs is critical to modern IT services, it is becoming significantly more challenging to perform log analysis at scale such that it provides actionable insight. The foremost reason is the sheer volume of log data. Software architecture paradigms such as containerization [7], micro-services [8,9], and serverless computing [10] push towards larger number of smaller components each generating their own log streams. Even with powerful search functions, text pattern matching, and aggregation tools, system operators are quickly overwhelmed with the volume and complexity. In addition, with the variety and diversity of cloud software, logs generated from the software and systems tend to have very different formats, levels of detail, and content.

Various approaches have been proposed in this field for diagnosing service failures using logs. A big proportion of research focuses on detecting anomalies or outliers from the logs [11–17] after a failure occurs. Log analysis has been a good target for applying data mining and AI techniques [18]. Although a large number of techniques are available, they detect anomalies via post-mortem analysis

or provide only statistical analysis of the monitored data, rather than revealing the actual importance and relevance in a cloud operational context. Therefore, it still remains a challenge for operators to be able to quickly narrow down to the small set of relevant logs for closer inspection.

Furthermore, cloud environments involve changes to applications, system software, patches, cluster provisioning, tenancy, and configuration. In a practical setting, even detected anomalies and outliers, while statistically relevant as rare occurrences, may be normal in a complex cloud setting based on the operational context (for example, uncommon but normal load changes, software updates and patches, routine configuration modifications, system utilization changes). With increased use of monitoring tools and analysis technologies to tame complexity, false alarms, and particularly excess alarm fatigue for system administrators is increasing. This leads to system administrators being overwhelmed by the number of anomalies reported and begin to ignore many of them. Further, the diversity of log types adds another layer of challenge for developers. Therefore, it is key that such analytics or AI identified statistical outliers be filtered or at least sorted in order of importance in the cloud operational domain context. An ideal log based alerting system would not only look at correlations between logs to predict outliers, but also temporal correlations between active system operations (such as planned configuration changes, planned or unplanned maintenance schedules, load patterns, daily health check runs, white lists, etc.). Incorporating and correlating this operational domain knowledge with AI and analysis is the ideal goal.

To this end, we have designed and implemented a novel method, called Priolog, to narrow down from volumes of raw logs to the small number of most relevant logs that are highly likely to carry the key information to the root cause of the problem. At the high-level, Priolog applies three independent analyses—log template temporal correlation analysis, log template frequency analysis and term frequency analysis. In the first template temporal correlation analysis, we look at the correlations among the time-series of log message types in order to find outlying log message types. A log message type, or a *log template*, is a static string part of a log message within which contextual values or strings are embedded to reflect the current execution state. We transform the raw log streams into *n* time-series, one from each log template, and cluster them by strength of temporal correlations. Intuitively, log message types that do not cluster well with others are a product of abnormal behaviors likely to contain important information. Such log templates are given high scores in this analysis. In the second template frequency analysis, we look at the frequency of the log messages per their corresponding log message types and identify the ones that significantly departs from the *normal level of frequencies*. If certain log templates show sudden change of frequencies, this may be indicative of unusual activities. Investigation of them may be helpful to finding the root cause. Similarly a newly appearing log template types (i.e., increase of frequency from 0 to some value) could carry a high-value information. The third term frequency (TF) analysis tries to compute the scores of individual messages by taking into account the rareness of individual words within the log messages. The score of a log message is computed as a function of the rareness scores of individual component words. The reasoning behind this is that certain words that are not seen in other messages could be a direct description of abnormal conditions. This 3rd analysis step is intended to further narrow down the log templates having similar score from previous two analyses into the ones of higher value. As a final output, Priolog generates a ranked list of log message types sorted by the product of all three ranks from the analyses.

In order to verify the effectiveness of our methodology in real-world problems, we have applied Priolog into the problem determination task for OpenStack [19], the open-source IaaS (Infrastructure-as-a-Service) platform. We first created several failure scenarios of selected operations—launch of oversized VM (Virtual Machine), VM launch failure due to core component failure, and exceeding the VM volume attach limit. For each failure cases, Priolog was able to successfully list highly relevant logs containing direct hints to the within top-ten of the ranked list of log message types.

The objective of Priolog is to support the problem diagnosis and root cause analysis by bringing upfront the most important logs to the user. In this regards, our contributions are: (i) Proposing novel

log selection algorithm made of three independent analyses, and (ii) demonstration of the feasibility through evaluation of popular software. From (i) we learn that simple single criteria does not work well in searching the most important logs, but the combination of multiple techniques must be applied to obtain reasonable accuracy. From (ii) we find that application logs do contain a wealth of useful information hidden in the logs. One requirement for `Priolog` to be effective was that the logs should contain high-value information in the first place. Through evaluation, we verify that it is the case. This motivates us to develop more advanced techniques to further identify relevant information from logs.

The rest of the paper is organized as follows. Section 2 provides details of architecture designs and justifications. Section 3 presents our evaluation of the effectiveness of Priolog using OpenStack. Related work is described in Section 4. Finally, we provide a concluding remark in Section 5.

## 2. Design of Priolog

The overall architecture of Priolog is shown in Figure 1. The input data to the Priolog are:

- Log template list: List of log templates prepared by the log template discovery algorithm. The technique of log template discovery is out-of-scope. We assume this list is made available by using existing techniques. Log template list is used in the (a) Log Template Time Series Analysis and (b) Log Template Frequency Analysis stages.
- Normal logs: logs collected from normal and error-free execution of the application of interest. This is used only for the (b) Log Template Frequency Analysis stage to build the log template frequency vector.
- Target logs: logs collected from the application instance that experiences some problem. This is the main input data from which we are trying to determine the root cause of the reported problem.

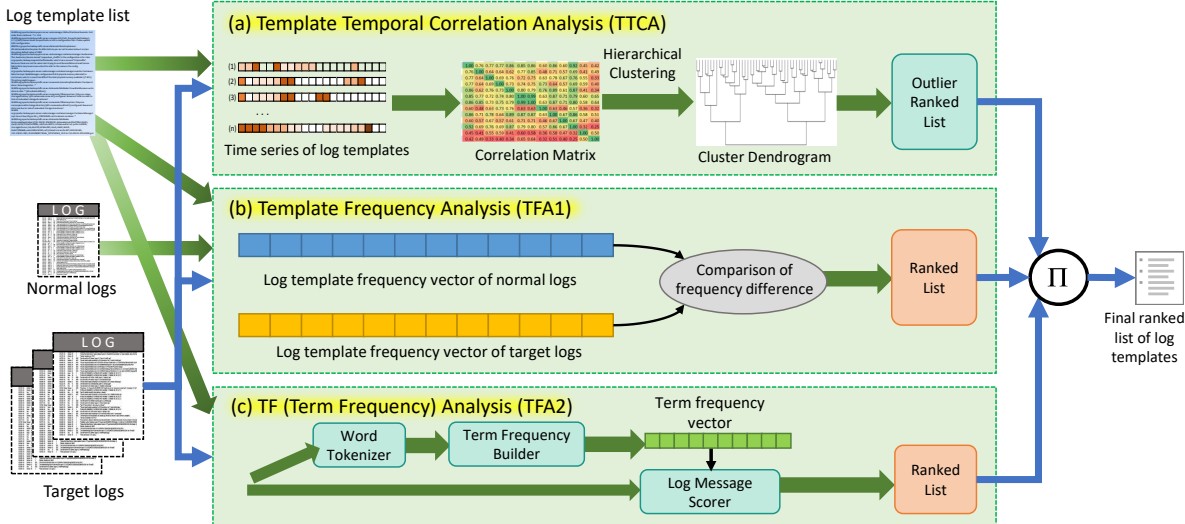

**Figure 1.** Priolog Architecture.

The log processing flow in the Priolog consists of three separate stages—(a) template temporal correlation analysis (TTCA), (b) template frequency analysis (TFA1), and (c) term frequency analysis (TFA2). Each analysis produces its own ranked list of log templates. Assume that there is a list of log templates of size $n$ for an application. Let us denote $p_i$ as the $i$th log template in this list. Also, the function $r()$ returns the rank of a log template in the ranked list. That is, $r_{TTCA}(p_i)$ would be equals to some integer which represents the rank in the list generated by the template temporal correlation analysis (TTCA). New final score for a log template $p_i$ is currently generated by calculating the product of all these three ranks.

$$R_i = \prod_{j}^{TTCA,TFA1,TFA2} r_j(p_i) \tag{1}$$

where $TTCA, TFA1, TFA2$ are the acronyms of three analyses. The final list of log templates is sorted by this metric $R_i$. We have chosen the product instead of summation or average of ranks because we wanted to penalize more if rank of any one of the three analyses results low. Summation or average of ranks are unable to differentiate the log templates that ranks consistently from the ones that has high variations across the analysis.

*2.1. Template Temporal Correlation Analysis*

The goal of this analysis is to identify a set of log templates that are unrelated to the major activities in the application. We refer to such log templates as outliers. The reasoning behind this analysis is as follows. All activities of applications can be categories as two kinds: (i) periodic and automatic background jobs and (ii) predefined set of operations triggered by requests from other components or users. Therefore, applications tend to generate log messages in some fixed and repeated sequences depending on the combination of such activities. Viewing the logs along the time progression, logs will be in a burst of several log messages as a consequence of executing certain tasks. If we aggregate the logs and count the log templates, these co-occurring log templates will maintain certain ratios. In addition, these logs from the log templates will be located within the time range. The overall logs we see in the log files are the interleaving of many such activities.

For the problem diagnosis, we are interested in locating the logs of high importance that are not part of any common activities within the applications. Thus, we want to filter out the group of logs whose log templates have high temporal correlations within themselves. Such temporally correlated logs are probably the output from normal and uninteresting activities. If the application encounters error conditions or problems, it will start to execute error-handling logic. This will be reflected in the logs as the a stream of log sequences that are previously unseen. This, in turn, implies that there will be logs from new (or rare) log templates and they would not be temporally correlated to the existing log templates or background jobs which were already going on within the application.

In order to take advantage of this principle in narrowing down the logs to the most important ones, we first convert the logs as multiple time series data per each log template. Then, we try to identify the log templates that have high temporal correlations and cluster them together. At the end of this task, if there are some log templates that do not belong to any clusters, these would be treated as outliers.

2.1.1. Log Templates

Log templates are a finite set of static string patterns that are used as a template from which actual logs are produced by embedding values of state variables or numbers that represent the current execution state of the application. They are usually the hard-coded part of the string within the log print statements of the application code. Table 1 are some of the example log templates found in the OpenStack platform. The variable parts within the log templates are expressed as wildcard following the regular expression notation.

Although Table 1 lists only 10 log templates, the length of full list can be at the order of hundreds or thousands. The frequency distribution of log templates exhibit power law pattern which implies that majority of the log message are from small subset of the entire log templates. Also, there are large number of log templates that are used in small numbers or infrequently. Some of the log templates are not seen until unusual conditions arise during the application run.

**Table 1.** Example log templates of OpenStack. Variable parts are denoted by the wildcard character(*).

| ID | Log Template |
|----|--------------|
| 1 | DEBUG * [None * None None] Agent rpc_loop - iteration: * started {{(pid=*) rpc_loop *}} |
| 2 | DEBUG * [None * None None] Agent rpc_loop - iteration: * completed. Processed ports statistics: {'regular': {'updated': * 'added': * 'removed': * Elapsed:* {{(pid=*) loop_count_and_wait *}} |
| 3 | DEBUG oslo_service.periodic_task [None * None None] Running periodic task * {{(pid=*) run_periodic_tasks *}} |
| 4 | DEBUG neutron_lib.callbacks.manager [None * None None] Notify callbacks * for agent, after_update {{(pid=*) _notify_loop *}} |
| 5 | DEBUG oslo_service.service [None * None None] * = * {{(pid=*) log_opt_values *}} |
| 6 | DEBUG nova.api.openstack.wsgi_app [-] * = * {{(pid=*) log_opt_values .*}} |
| 7 | ERROR cinder.service [-] Manager for service cinder-volume * is reporting problems, not sending heartbeat. Service will appear "down". |
| 8 | DEBUG oslo_concurrency.lockutils [-] Lock "_check_child_processes" released by "neutron.agent.linux.external_process._check_child_processes" :: held * {{(pid=*) inner *}} |
| 9 | AUDIT nova.compute.claims [* admin demo] instance: * Total Disk: * GB, used: * GB |
| 10 | DEBUG nova.openstack.common.rpc.amqp [* admin demo] MSG_ID is * multicall * |

In this work we assume that we have already prepared such set of log templates for a given application using existing techniques. Accurately discovering the log templates from the given set of log data is an active field of research and there are several techniques available for us to use [20–23]. It is not the goal of this work to design new log template discovery techniques.

2.1.2. Log Template Time-Series Generation

As a first step of the temporal correlation analysis, Priolog takes in the target logs and convert them into $n$ time series assuming the length of log template list is $n$. For the time series generation, we define a time window $\rho$ and count the number of logs appearing within each window for each log template. The window size is determined by dividing the duration of log start and end time by 50 for convenience. This factor can be adjusted as necessary to some other value. In our cases, it ranges roughly from 10 to 100 ms.

The visualization of sample time series obtained from the actual logs is shown in Figure 2. Since there are three axes (log template IDs, time, and log counts), we used a heat-map style for the log count quantity. Each small square indicates the presence of logs and the color intensity is the relative density. If the color is stronger toward red, it implies there are large number of logs within that time window. The log template ID is roughly sorted in a way that smaller ID is assigned to more frequently used log templates. The log template ID 0 is a special template where unclassified logs are assigned. Visualization reveals that there are several group of log templates that has temporal locality. As the log template ID increases (i.e., to the right side of the figure), the log counts become scarce and most of the time window is blank.

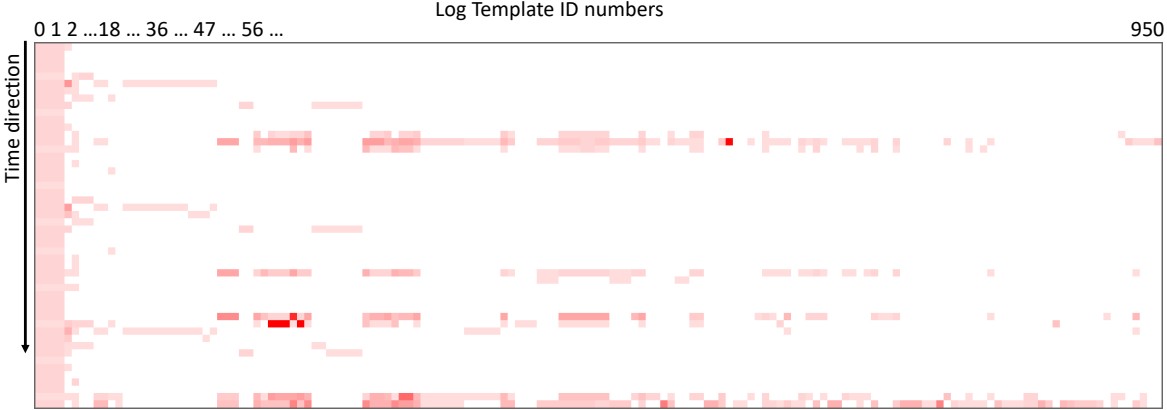

**Figure 2.** Heatmap of example time series of log templates.

### 2.1.3. Correlation Matrix Construction

For all pairs of time series derived from the log template counts, we calculate the correlation coefficients and construct $n \times n$ correlation matrix. In computing the correlation we apply the Pearson correlation coefficient metric which is defined as the covariance of two time series data divided by the product of two standard deviations. The value closer to 1 indicates stronger correlation.

We perform this step in order to initialize the distances between log template time series for the following hierarchical clustering step. As the hierarchical clustering progresses, this correlation matrix will have new time series added and the correlation coefficients calculated as needed. More details are explained in the following subsection where hierarchical clustering step is explained.

Intuitively, strong correlation of log template time series is interpreted as coming from the same system activities. Certain tasks of applications tend to generate logs only from certain subset of log templates since the application logic will execute identical log print statements for a given task although there may be minor non-deterministic variations. This behavior will, in turn, generate temporally correlated series of log template counts which we aim to discover by clustering log templates time series.

### 2.1.4. Hierarchical Clustering

Using the initial correlation coefficient matrix from the previous step, Priolog now performs a hierarchical clustering. Using the bottom-up fashion, we repeatedly cluster two log template time series $\tau_i$ and $\tau_j$ that have highest correlation coefficient among the current set of clusters. These two log template time series are, then, merged into one denser time series $\tau_k$ and put back into the clustering. New entries are added to the correlation coefficient matrix. At the same time, $\tau_i$ and $\tau_j$ are removed from the matrix making the overall size of the matrix shrink by 1 in both dimensions. The Pearson correlation is calculated to populate this new entry in the matrix. Then, next highest correlation values are sought among the remaining log templates and the merged groups. This process of merging repeats until we end up with one final cluster. These processes are described in Algorithm 1. Figure 3a shows the result of hierarchical clustering as a dendrogram.

---

**Algorithm 1:** Hierarchical clustering in Priolog.

---

1 **Input:**Correlation coefficient matrix *C*;
2 **Output:**Dendrogram *D*;
3 *D* = {}; /* *empty dendrogram* */
4 num_time_series=dimension_of(*C*);
5 **while** *num_time_series>2* **do**
6 　　$\tau_1,\tau_2$=PickTwoHighestCorrelation(*C*);
7 　　$\tau_3$ = MergeTimeSeries($\tau_1,\tau_2$);
8 　　*C* = RemoveTimeSeries(*C*,$\tau_1$);
9 　　*C* = RemoveTimeSeries(*C*,$\tau_2$);
10 　　*C* = AddTimeSeries(*C*,$\tau_3$);
11 　　UpdateMatrix(*C*);
12 　　AddNewEdge(*D*,$\tau_1,\tau_2$)
13 　　UpdateCluster(*D*,$\tau_3$)
14 　　num_time_series–;
15 **end**

---

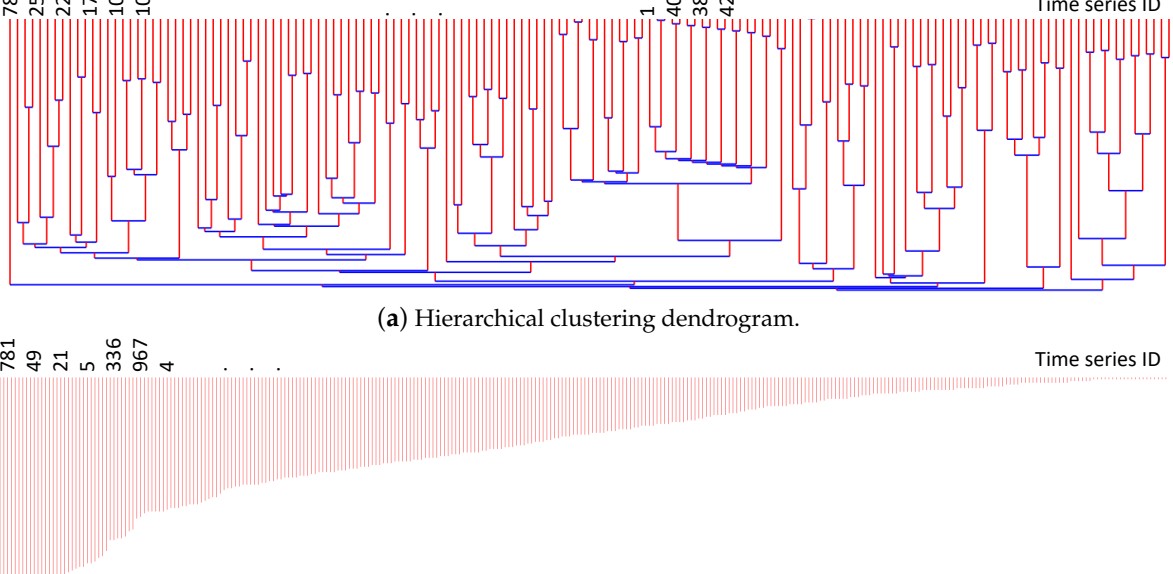

(**a**) Hierarchical clustering dendrogram.

(**b**) Sorted by vertical line length: duration of time the cluster remained unclustered.

**Figure 3.** Hierarchical clustering performed by Priolog.

### 2.1.5. Outlier Identification

The construction of the hierarchical clustering structure does not yet provide us with the quantification of a logs' importance. In order to obtain this, we use the time duration of the time series that remained un-merged for the longest time. If a log template time series has a low correlation with any other log templates, it will remain un-selected by the clustering algorithm for a long time and be merged towards the end of the clustering process. This implies that such log templates are the product of some unusually outlying activity in the system. If some logs are generated from natural normal behaviors, they would show higher correlation with several other co-occurring log templates. Note that the outlier can be a single log template or a group of small log templates that are merged early on during the clustering and remained unmerged for a long time. Our goal is to find such outlying log template (groups) and highlight them as important or 'requiring attention'.

Figure 3b is the sorted list of vertical edges from (a). The left-most vertical line comes from the time series 781 in the figure and this is the most uncorrelated time series (and the log templates that comprises this time series). Note that the time series could be the one created by merging several log templates together during the clustering process. In such case, it means that those group of log templates are together part of the same activity, but they, as a group, is distinctly unique compared to other activities. The final rank of log templates as the result of the analysis is assigned using this sorted list of vertical lines.

## 2.2. Template Frequency Analysis

The approach of this analysis is to score the log templates based on the observed frequencies. The assumption behind this analysis is that, if the number of logs of certain log templates shows higher frequency than normal, this log is probably important. This idea is first proposed in the work by Sabato et al. [24]. The key to the successful application of this idea is how to define the 'normal' level of frequency for each log templates. We adopt this method in Priolog as one of the analysis.

### 2.2.1. Frequency Vector Construction

When it was first proposed, Sabato et al. [24] assumed an environment where there were large number of computers working on heterogeneous tasks. Thus, one of the focus of this work was to apply the clustering to build accurate cluster of computers performing similar tasks. The 'normality' is, then, defined per these clusters. However, in our settings, we do not assume the existence of large number of servers. Instead, we assume to maintain a staging environment identical (or similar) to the production settings and we perform planned experiments to collect the logs that represent the 'normal' runs. Nevertheless the output of this is the same—probability vector $P = \{p_1, p_2, ..., p_n\}$. The probability vector $P$ is of length $n$, the number of log templates, and it holds the probability of corresponding log templates to appear in the logs.

Given the target logs, Priolog builds the frequency vector $F$ similar to the probability vector. This frequency vector is normalized so that the sum of all the elements would be 1 and then compared with $P$. Then, we compute the ratio between $F_i$ and $P_i$ as $|F_i/P_i|$ for each log template $i$. These ratios of log templates are used to rank them. If a log template is previously unknown, it will not be in $P_i$. In such case, the ratio will be infinity and that log template will have the rank 1 since newly appearing logs are likely to be important.

## 2.3. Term Frequency (TF) Analysis

The first two analyses are effective for ruling out unimportant logs to reduce the log volumes. Unfortunately, they are not fit for pinpointing the logs that contain high-value information because of these reasons. For the template temporal correlation analysis (TTCA), in order for this analysis to work well, the time series data must be 'rich' with data. However, for large number of unpopular log templates, their time series is mostly filled with zeros and very occasionally (or never) non-zeros. Such times series does not generate meaningful correlation coefficients. This implies that the clustering by the temporal information will only be able to identify clusters that are uninteresting to us that should be eliminated. Most of the times, high-value logs are usually from the rarely occurring log templates. For the template frequency analysis (TFA1), the problem with this is that the increase of log template frequency does not strongly indicate that such logs are more important. Rather, increased log instances can be considered as less valuable because it is more common. Another problem is that there can be excessively large number of log templates that suddenly appear which were non-existing in the normal case. We have experienced this problem frequently and were unable to further narrow down the logs to the most relevant ones.

The third term frequency analysis (TFA2) is introduced to address these shortcomings of other analyses. The hypothesis is that, among the candidate logs, if any one of them contains the words that do not appear often in other log templates, then probably such words are important. It may be the

words that directly describe the abnormal condition because the programmer may have added log print statements in the code in such a way that current conditions are revealed. Based on this idea, we perform a word-level frequency analysis as the third analysis.

A log template contains many tokens that include numbers, special characters, and punctuation characters. In this analysis, we mean by words only those tokens that contain alphabets and underscore. The score for the rank, $r_i$, is calculated as Equation (2). Let us use $S_i$ to denote the total word set in the $i$th log template. The total number of effective words in the $i$th log template is $|S_i|$. Let $f_w$ be the frequency of word $w$ in the entire log template set. Then, $f_w / \sum_{w'} f_{w'}$ becomes the overall proportion at the global level of $w$ in the log template set. In words, we calculate the average of proportions of all the words within a given log template and use them as the score.

$$r_i = \frac{1}{|S_i|} \sum_{j \in S_i} \left( \frac{f_{w_j}}{\sum_{w'} f_{w'}} \right) \tag{2}$$

Final ranked list of this analysis is generated by sorting the log templates by this $r_i$ scores.

## 3. Evaluation

In order to verify the effectiveness, we have applied Priolog to the OpenStack platform [19]. OpenStack is a popular open-source Infrastructure-as-a-Service (IaaS) platform that was launched in 2010. Since then it has grown to contain more than 30 component groups each comprising several sub-components. We have created following four failure scenarios in the OpenStack Stein release, installed with DevStack.

(1)　Component (`nova-compute`) failure: In this scenario, we simulate the component failure by killing one of the core components, `nova-compute`. Then, we try to launch a VM.
(2)　Component (`neutron-dhcp-agent`) failure: Similar to the first case, we kill the `neutron-dhcp-agent` component and observe the logs.
(3)　Oversized VM launch failure due to insufficient resource: In this scenario, we try to create a VM instance that requires too much memory so that current system cannot handle it.
(4)　Volume max limit exceeded: OpenStack VM has default volume count limit set to 10. We try to add the 11th volume and observe the failure.

Each case has been repeated at least twice (with case (3) done three times) giving us total of nine test results. Out of these nine cases, we present detailed results of three cases in the following subsections. We also provide summary of all nine experiments in Section 3.4.

### 3.1. Component Failure

In this scenario, we intentionally terminated the `nova-compute` component of the OpenStack. The `nova-compute` is responsible for performing the actual tasks of launching a requested VM. It simulates the case where one of the core OpenStack component silently fails and the admin has no clue as to why VM suddenly fails to launch. When we press the Launch button on the Horizon UI, it runs for a while and produces a message that does not related to the failure of the `nova-compute` component.

Figure 4 presents the analyses results of Priolog. Figure 4a–c are the ranking results of the TTCA, TFA1, and TFA2, respectively. As a final rank, we see that the log template number 951 is ranked at No. 1. This log template 951 contains decisive information: "Seems service `nova-compute` on host * is down." This log template is consistently ranked high in the TTCA and TFA1. The rank is somewhat low in the TFA2, but the overall rank obtained by product of all three ranks came out as No. 1. This shows that Priolog was able to find the key information from the huge volume of logs successfully.

**Template Temporal Correlation Analysis (Hierarchical Clustering)**

| Rank | Tmpl # | Cluster Time | Log Template Message |
|---|---|---|---|
| #1 | 21 | 188 | DEBUG neutron.db.agents_db \[None .* None None\] Agent healthcheck: found .* active agents {{\(pid=.*\) agent_health_check .*}} |
| #2 | 4 | 186 | DEBUG oslo_service.periodic_task \[None .* None None\] Running periodic task .* {{\(pid=.*\) run_periodic_tasks .*}} |
| #3 | 954 | 184 (0) | DEBUG .* \[None .* None None\] ofctl request version=.*,msg_type=.*,msg_len=.*,xid=.*,OFPFlowStatsRequest\(cookie=.*,cookie_mask=.*,flags=.*,match=.*\(oxm_fields=(.*)\),out_group=.*,out_port=.*,table_id=.*,type=.*\) result \[OFPFlowStatsReply\(body=\[OFPFlowStats\(byte_count=.*,cookie=.*,duration_nsec=.*,duration_sec=.*,flags=.*,hard_timeout=.*,idle_timeout=.*,instructions=\[.*\],length=.*,match=.*\(oxm_fields=(.*)\),packet_count=.*,priority=.*,table_id=.*\)\],flags=.*,type=.*\)\] {{\(pid=.*\)_send_msg.*}} |
| | | 184 (1) | DEBUG neutron.agent.linux.utils \[None .* None None\] Found cmdline \['ovsdb\-client', 'monitor', 'tcp:.*', 'Interface', 'name,ofport,external_ids', '\-\-format=.*'\] for process with PID .* {{\(pid=.*\) get_cmdline_from_pid .*}} |
| #4 | 5 | 180 | DEBUG neutron_lib.callbacks.manager \[None .* None None\] Notify callbacks .* for agent, after_update {{\(pid=.*\) _notify_loop .*}} |
| #5 | 951 | 179 | DEBUG nova.servicegroup.drivers.db \[.* .* .* .*\] Seems service nova-compute on host .* is down. Last heartbeat was .*. Elapsed time is .* |
| #6 | 955 | 176 (9) | DEBUG oslo_concurrency.lockutils \[\-\] Lock "_check_child_processes" released by " neutron.agent.linux.external_process._check_child_processes" :: held .* {{\(pid=.*\) inner .*}} |
| | | 176 (10) | DEBUG oslo_concurrency.lockutils \[\-\] Lock "_check_child_processes" acquired by " neutron.agent.linux.external_process._check_child_processes" :: waited .* {{\(pid=.*\) inner .*}} |
| #7 | 189 | 175 | DEBUG oslo_db.sqlalchemy.engines \[None .* None None\] MySQL server mode set to STRICT_TRANS_TABLES,STRICT_ALL_TABLES,NO_ZERO_IN_DATE,NO_ZERO_DATE,ERROR_FOR_DIVISION_BY_ZERO,TRADITIONAL,NO_AUTO_CREATE_USER,NO_ENGINE_SUBSTITUTION {{\(pid=.*\) _check_effective_sql_mode .*}} |

(**a**) Template Temporal Correlation Analysis

**Ratio Differences of Log Messages**

| Rank | Tmpl # | Score | Log Template Message |
|---|---|---|---|
| #6 | 413 | ∞ | DEBUG oslo_concurrency.lockutils \[None .* None None\] Lock "event\-dispatch" acquired by .* :: waited .* {{\(pid=.*\) inner .*}} |
| #7 | 553 | ∞ | DEBUG oslo_concurrency.processutils \[None .* demo admin\] Running cmd \(subprocess\): .* \-m oslo_concurrency.prlimit \-\-as=.* \-\-cpu=.* \-\- env LC_ALL=.* LANG=.* qemu\-img info .* \-\-force\-share {{\(pid=.*\) execute .*}} |
| #8 | 554 | ∞ | DEBUG oslo_concurrency.processutils \[None .* demo admin\] CMD .* \-m oslo_concurrency.prlimit \-\-as=.* \-\-cpu=.* \-\- env LC_ALL=.* LANG=.* qemu\-img info .* \-\-force\-share" returned: .* in .* {{\(pid=.*\) execute .*}} |
| #9 | 951 | ∞ | DEBUG nova.servicegroup.drivers.db \[.* .* .* .*\] Seems service nova-compute on host .* is down. Last heartbeat was .*. Elapsed time is .* |
| #10 | 225 | 8.4091925 4658 | DEBUG nova.compute.multi_cell_list \[None .* demo admin\] Listed batch of .* results from cell out of .* limit. Returned .* total so far. {{\(pid=.*\) do_query .*}} |
| #11 | 162 | 7.2624844 7205 | DEBUG keystone.server.flask.request_processing.middleware.auth_context \[None .* None admin\] Validating token access rules against request {{\(pid=.*\) validate_allowed_request .*}} |
| #12 | 163 | 7.2624844 7205 | DEBUG keystone.server.flask.request_processing.middleware.auth_context \[None .* None admin\] Authenticating user token {{\(pid=.*\) process_request .*}} |
| #13 | 231 | 6.6254244 3064 | DEBUG neutron.pecan_wsgi.hooks.policy_enforcement \[None .* demo admin\] Attributes excluded by policy engine: \[u'shared'\] {{\(pid=.*\) _exclude_attributes_by_policy .*}} |
| #14 | 214 | 4.9690683 2298 | DEBUG neutron.pecan_wsgi.hooks.policy_enforcement \[None .* demo admin\] Attributes excluded by policy: \[u'vlan_transparent'\] {{\(pid=.*\) _exclude_attributes_by_policy .*}} |
| #15 | 228 | 4.5868322 9814 | DEBUG nova.compute.api \[None .* demo admin\] Searching by: {'deleted': False, u'project_id': .* {{\(pid=.*\) get_all .*}} |

(**b**) Template Frequency Analysis

**Term Frequency Analysis**

| Rank | Tmpl # | Frequency | Log Template Message |
|---|---|---|---|
| #49 | 54 | 3.25 | DEBUG cinder.api.openstack.wsgi \[None .* demo admin\] Empty body provided in request {{\(pid=.*\) get_body .*}} |
| | | | ['empty', 'body', 'provided', 'request'] |
| #50 | 951 | 3.2857 1428571 | DEBUG nova.servicegroup.drivers.db \[.* .* .* .*\] Seems service nova-compute on host .* is down. Last heartbeat was .*. Elapsed time is .* |
| | | | ['seems', 'service', 'host', 'last', 'heartbeat', 'elapsed', 'time'] |
| #51 | 68 | 3.33 | DEBUG glance.api.middleware.version_negotiation \[None .* demo admin\] Determining version of request: GET .* |
| | | | ['determining', 'version', 'get'] |
| #52 | 480 | 3.33 | DEBUG neutron.agent.securitygroups_rpc \[None .* None None\] Refreshing firewall for .* devices {{\(pid=.*\) setup_port_filters .*}} |
| | | | ['refreshing', 'firewall', 'devices'] |
| #53 | 53 | 3.5 | INFO cinder.api.openstack.wsgi \[None .* demo admin\] .* returned with HTTP .* |
| | | | ['returned', 'http'] |
| #54 | 191 | 3.5 | DEBUG neutron.wsgi \[None .* demo admin\] .* returned with HTTP .* {{\(pid=.*\)_call_.*}} |
| | | | ['returned', 'http'] |
| #55 | 613 | 3.5 | DEBUG neutron.agent.linux.dhcp \[\-\] Building host file: .* {{\(pid=.*\) _output_hosts_file .*}} |
| | | | ['building', 'host'] |

(**c**) Term Frequency Analysis

**Final Ranking**

| Rank | Tmpl # | Score | Log Template Message |
|---|---|---|---|
| #1 | 951 | 250 | DEBUG nova.servicegroup.drivers.db \[.* .* .* .*\] Seems service nova-compute on host .* is down. Last heartbeat was .*. Elapsed time is .* |
| #2 | 228 | 735 | DEBUG nova.compute.api \[None .* demo admin\] Searching by: {'deleted': False, u' project_id': .* {{\(pid=.*\) get_all .*}} |
| #3 | 193 | 1950 | DEBUG neutron.quota.resource \[None .* demo admin\] Usage tracker for resource:.* and tenant:.* is out of sync, need to count used quota {{\(pid=.*\) count_used .*}} |
| #4 | 59 | 2867 | DEBUG keystone.common.fernet_utils \[None .* Loaded .* Fernet keys from .* but `.* [fernet_tokens]` max_active_keys = .* perhaps there have not been enough key rotations to reach `max_active_keys` yet`? {{\(pid=.*\) load_keys .*}} |
| #5 | 194 | 3000 | DEBUG neutron.quota.resource \[None .* demo admin\] Quota usage for .* was recalculated. Used quota:.* {{\(pid=.*\) count_used .*}} |
| #6 | 21 | 3003 | DEBUG neutron.db.agents_db \[None .* None None\] Agent healthcheck: found .* active agents {{\(pid=.*\) agent_health_check .*}} |
| #7 | 66 | 3060 | DEBUG glance.api.middleware.version_negotiation \[None .* demo admin\] Matched version: .* {{\(pid=.*\) process_request .*}} |
| #8 | 200 | 3402 | INFO neutron.wsgi .* demo admin\] .* "GET /v.*/security-groups\?id=.* status: .* l en: .* time: .* |
| #9 | 39 | 3486 | DEBUG nova.compute.resource_tracker \[None .* None None\] Instance .* actively managed on this compute host and has allocations in placement: {u'resources': {u'VCPU': .* u'MEMORY_MB': .* u'DISK_GB': .* {{\(pid=.*\) _remove_deleted_instances_allocations .*}} |

(**d**) Final Rank Score

**Figure 4.** Ranking scores of `nova-compute` component failure case.

## 3.2. Launch of Oversized VM

In this scenario, we try to launch a large VM that is beyond the resource capacity of the host. The error message on the Horizon web interface is misleading as well in this case. It simply says it failed to launch an instance and asks to try it again later. Figure 5 is the result of analyses. The log template we are looking for in this failure case is the log template 949. This log contains the words "insufficient resource", which is a direct description of the problem at hand. Note that this log is in the INFO log level. Thus, simple keyword search of ERROR through the logs files will not lead to such information. In our experiment, Priolog ranked this as No. 3 which is within top 10 of the final rank.

**(a) Template Temporal Correlation Analysis**

Template Temporal Correlation Analysis (Hierarchical Clustering)

| Rank | Tmpl # | Cluster Time | | Log Template Message |
|---|---|---|---|---|
| #64 | 1025 | 61 | 26 | DEBUG oslo_concurrency.lockutils \[None .* None None\] Releasing lock .* {{\(pid=.*\) lock .*}} |
| | | | 412 | DEBUG oslo_concurrency.lockutils \[None .* None None\] Lock .* released by "nova .context.get_or_set_cached_cell_and_set_connections" :: held .* {{\(pid=.*\) inner .*}} |
| | | | 414 | DEBUG oslo_concurrency.lockutils \[None .* None None\] Lock .* acquired by "nova .context.get_or_set_cached_cell_and_set_connections" :: waited .* {{\(pid=.*\) inner .*}} |
| | | | 708 | INFO nova.compute.rpcapi \[None .* None None\] Automatically selected compute RPC version .* from minimum service version .* |
| #65 | 196 | 60 | | DEBUG keystone.server.flask.request_processing.middleware.auth_context \[None .* demo admin\] Authenticating user token {{\(pid=.*\) process_request .*}} |
| #66 | 944 | 60 | | DEBUG keystone.server.flask.request_processing.middleware.auth_context \[None .* demo admin\] Validating token access rules against request {{\(pid=.*\) validate_allowed_request .*}} |
| #67 | 951 | 60 | 172 | ERROR nova.conductor.manager .* |
| | | | 949 | INFO nova.scheduler.manager \[.* .* .* .*\] Got no allocation candidates from the Placement API. This could be due to insufficient resources or a temporary occurrence as compute nodes start up. |
| #68 | 952 | 60 | 158 | DEBUG oslo.privsep.daemon \[\-\] privsep: .* {{\(pid=.*\) _call_back .* |
| | | | 174 | DEBUG neutron_lib.callbacks.manager \[None .* None None\] Notify callbacks .* for port, provisioning_complete {{\(pid=.*\) _notify_loop .*}} |
| #69 | 953 | 60 | 168 | DEBUG oslo_concurrency.lockutils \[\-\] Lock "privileged\-ip\-lib" released by "neutron.privileged.agent.linux_ip_lib.get_link_devices" :: held .* {{\(pid=.*\) inner .*}} |
| | | | 175 | DEBUG neutron.db.provisioning_blocks \[None .* None None\] Provisioning complete for port .* triggered by entity .* {{\(pid=.*\) provisioning_complete .*}} |

**(b) Template Frequency Analysis**

Ratio Differences of Log Messages

| Rank | Tmpl # | Score | Log Template Message |
|---|---|---|---|
| #12 | 490 | ∞ | DEBUG glance.db.sqlalchemy.metadef_api.namespace \[None .* demo admin\] context.is_admin=.*; context.owner=.* {{\(pid=.*\) _select_namespaces_query .*}} |
| #13 | 553 | ∞ | DEBUG oslo_concurrency.processutils \[None .* demo admin\] Running cmd \(subprocess\): .* \-m oslo_concurrency.prlimit \-\-as=.* \-\-cpu=.* \-\- env LC_ALL=.* LANG=.* qemu\-img info .* \-\-force\-share {{\(pid=.*\) execute .*}} |
| #14 | 554 | ∞ | DEBUG oslo_concurrency.processutils \[None .* demo admin\] CMD .* \-m oslo_concurrency.prlimit \-\-as=.* \-\-cpu=.* \-\- env LC_ALL=.* LANG=.* qemu\-img info .* \-\-force\-share" returned: .* in .* {{\(pid=.*\) execute .*}} |
| #15 | 637 | ∞ | DEBUG cinder.volume.api \[None .* demo admin\] Could not evaluate value available, assuming string {{\(pid=.*\) check_volume_filters .*}} |
| #16 | 949 | ∞ | INFO nova.scheduler.manager \[.* .* .* .*\] Got no allocation candidates from the Placement API. This could be due to insufficient resources or a temporary occurrence as compute nodes start up. |
| #17 | 191 | 6.5523850 | DEBUG neutron.wsgi \[None .* demo admin\] .* returned with HTTP .* {{\(pid=.*\) _call .*}} |
| #18 | 225 | 5.2419080 | DEBUG nova.compute.multi_cell_list \[None .* demo admin\] Listed batch of .* results from cell out of .* limit. Returned .* total so far. {{\(pid=.*\) do_query .*}} |
| #19 | 62 | 4.1498438 | DEBUG cinder.api.openstack.wsgi \[None .* demo admin\] Calling method 'all' {{\(pid=.*\) _process_stack .*}} |
| #20 | 53 | 3.3853989 | INFO cinder.api.openstack.wsgi \[None .* demo admin\] .* returned with HTTP .* |
| #21 | 54 | 3.3853989 | DEBUG cinder.api.openstack.wsgi \[None .* demo admin\] Empty body provided in request {{\(pid=.*\) get_body .*}} |
| #22 | 55 | 3.3853980 | INFO cinder.api.openstack.wsgi \[None .* demo admin\] GET .* |

**(c) Term Frequency Analysis**

Term Frequency Analysis

| Rank | Tmpl # | Frequency | Log Template Message |
|---|---|---|---|
| #10 | 317 | 1.0 | DEBUG oslo_db.sqlalchemy.engines \[None .* demo admin\] MySQL server mode set to STRICT_TRANS_TABLES,STRICT_ALL_TABLES,NO_ZERO_IN_DATE,NO_ZERO_DATE,ERROR_FOR_DIVISION_BY_ZERO,TRADITIONAL,NO_AUTO_CREATE_USER,NO_ENGINE_SUBSTITUTION {{\(pid=.*\) _check_effective_sql_mode .*}} |
| | | | ['mysql', 'server', 'mode', 'set', 'traditional'] |
| #11 | 432 | 1.0 | DEBUG neutron.agent.linux.dhcp \[\-\] Setting .* gateway for dhcp netns on net .* to .* {{\(pid=.*\) _set_default_route_ip_version .*}} |
| | | | ['setting', 'gateway', 'dhcp', 'netns', 'net'] |
| #12 | 78 | 1.25 | DEBUG cinder.manager \[None .* None None\] Notifying Schedulers of capabilities .. . {{\(pid=.*\) _publish_service_capabilities .*}} |
| | | | ['notifying', 'schedulers', 'capabilities', ''] |
| #13 | 949 | 1.285714 | INFO nova.scheduler.manager \[.* .* .* .*\] Got no allocation candidates from the Placement API. This could be due to insufficient resources or a temporary occurrence as compute nodes start up. |
| | | | ['got', 'allocation', 'candidates', 'placement', 'api', 'could', 'due', 'insufficient', 'resources', 'temporary', 'occurrence', 'compute', 'nodes', 'start'] |
| #14 | 637 | 1.33333 | DEBUG cinder.volume.api \[None .* demo admin\] Could not evaluate value available, assuming string {{\(pid=.*\) check_volume_filters .*}} |
| | | | ['could', 'evaluate', 'value', 'available', 'assuming', 'string'] |
| #15 | 175 | 1.4 | DEBUG neutron.db.provisioning_blocks \[None .* None None\] Provisioning complete for port .* triggered by entity .* {{\(pid=.*\) provisioning_complete .*}} |
| | | | ['provisioning', 'complete', 'port', 'triggered', 'entity'] |
| #16 | 225 | 1.625 | DEBUG nova.compute.multi_cell_list \[None .* demo admin\] Listed batch of .* results from cell out of .* limit. Returned .* total so far. {{\(pid=.*\) do_query .*}} |

**(d) Final Rank Score**

Final Ranking

| Rank | Tmpl # | Score | Log Template Message |
|---|---|---|---|
| #1 | 49 | 236 | DEBUG nova.compute.manager \[None .* None None\] CONF.reclaim_instance_interval <= .* skipping... {{\(pid=.*\) _reclaim_queued_deletes .*}} |
| #2 | 637 | 770 | DEBUG cinder.volume.api \[None .* demo admin\] Could not evaluate value available, assuming string {{\(pid=.*\) check_volume_filters .*}} |
| #3 | 949 | 845 | INFO nova.scheduler.manager \[.* .* .* .*\] Got no allocation candidates from the Placement API. This could be due to insufficient resources or a temporary occurrence as compute nodes start up. |
| #4 | 317 | 1058 | DEBUG oslo_db.sqlalchemy.engines \[None .* demo admin\] MySQL server mode set to STRICT_TRANS_TABLES,STRICT_ALL_TABLES,NO_ZERO_IN_DATE,NO_ZERO_DATE,ERROR_FOR_DIVISION_BY_ZERO,TRADITIONAL,NO_AUTO_CREATE_USER,NO_ENGINE_SUBSTITUTION {{\(pid=.*\) _check_effective_sql_mode .*}} |
| #5 | 235 | 1100 | DEBUG cinder.db.sqlalchemy.api \[None .* demo admin\] Building query based on filter {{\(pid=.*\) _process_snaps_filters .*}} |
| #6 | 65 | 1127 | DEBUG glance.api.middleware.version_negotiation \[None .* demo admin\] new path .* {{\(pid=.*\) process_request .*}} |
| #7 | 66 | 1127 | DEBUG glance.api.middleware.version_negotiation \[None .* demo admin\] Matched version: .* {{\(pid=.*\) process_request .*}} |
| #8 | 64 | 1173 | DEBUG glance.api.middleware.version_negotiation \[None .* demo admin\] Using url versioning {{\(pid=.*\) process_request .*}} |
| #9 | 19 | 1620 | DEBUG placement.requestlog .* service placement\] Starting request: .* "GET .* {{\(pid=.*\) __call__ .*}} |
| #10 | 79 | 1704 | DEBUG neutron.wsgi \[\-\] .* accepted .* {{\(pid=.*\) server .*}} |
| #11 | 3 | 1750 | DEBUG .* \[None .* None None\] Agent rpc_loop \- iteration:.* completed. Processed ports statistics: {'regular': {'updated': .* 'added': .* 'removed': .* Elapsed:.* |

**Figure 5.** Ranking scores of large (oversized) virtual machine (VM) instance launch failure case.

## 3.3. Exceeding Maximum Volume Count

This scenario shows the case where a user tries to attach additional storage volume to the VM instance and it fails due to predefined volume count limits. OpenStack defines various default resource limits per tenant (i.e., user account) such as the number of instances allowed, number of floating/fixed IP address allowed, number of security groups and the number of block storage volumes allowed per tenant. We have created 10 volumes and collected logs while trying to create the 11th volume. As can be seen in the Figure 6, the final ranked score list gives us the key log at the 5th rank.

**(a) Template Temporal Correlation Analysis**

| | Template Temporal Correlation Analysis (Hierarchical Clustering) | | |
|---|---|---|---|
| Rank | Tmpl # | Cluster Time | Log Template Message |
| #54 | 1082 | 56 | 37 DEBUG nova.scheduler.client.report \[None .* None None\] Inventory has not changed for provider .* based on inventory data: {u'VCPU': {u'allocation_ratio': .* u'total': .* u'reserved': .* u'max_unit': .* u'MEMORY_MB': {u'allocation_ratio': .* u'total': .* u'reserved': .* u'max_unit': .* u'DISK_GB': {u'allocation_ratio': .* u'total': .* u'reserved': .* u'step_size': .* u'min_unit': .* u'max_unit': .* {{\(pid=.*\) set_inventory_for_provider .*}} |
| | | | 38 DEBUG nova.compute.resource_tracker \[None .* None None\] Total usable vcpus: .* total allocated vcpus: .* {{\(pid=.*\) _report_final_resource_view .*}} |
| | | | 40 DEBUG nova.compute.resource_tracker \[None .* None None\] Final resource view: name=.* phys_ram=.* used_ram=.* phys_disk=.* used_disk=.* total_vcpus=.* used_vcpus=.* {{\(pid=.*\) _report_final_resource_view .*}} |
| | | | 41 DEBUG nova.compute.resource_tracker \[None .* None None\] Compute_service record updated for .* {{\(pid=.*\) _update_available_resource .*}} |
| | | | 42 DEBUG nova.compute.resource_tracker \[None .* None None\] Auditing locally available compute resources for .* \(node: .* {{\(pid=.*\) update_available_resource .*}} |
| | | | 43 DEBUG nova.compute.provider_tree \[None .* None None\] Inventory has not changed in ProviderTree for provider: .* {{\(pid=.*\) update_inventory .*}} |
| #55 | 956 | 55 | DEBUG nova.compute.utils \[.* .* .* .*\] \[instance: .*\] Build of instance .* aborted: VolumeLimitExceeded: Maximum number of volumes allowed \(.*\) exceeded for quota 'volumes'. .* |
| #56 | 781 | 54 | DEBUG nova.api.openstack.wsgi \[None .* demo admin\] Action: .* calling method:<="" td =""> |
| #57 | 950 | 54 | WARNING keystonemiddleware.auth_token \[.*\] A valid token was submitted as a service token, but it was not a valid service token. This is incorrect but backwards compatible behaviour. This will be removed in future releases. |

**(b) Template Frequency Analysis**

| | Ratio Differences of Log Messages | | |
|---|---|---|---|
| Rank | Tmpl # | Score | Log Template Message |
| #1 | 126 | ∞ | DEBUG nova.compute.manager \[None .* None None\] Cleaning up deleted instances with incomplete migration {{\(pid=.*\) _cleanup_incomplete_migrations .*}} |
| #2 | 946 | ∞ | DEBUG nova.api.openstack.compute.server_external_events \[.* service nova\] Unable to find a host for instance .* Dropping event .* {{\(pid=.*\) create .*}} |
| #3 | 947 | ∞ | INFO nova.api.openstack.wsgi \[.* service nova\] HTTP exception thrown: .* |
| #4 | 948 | ∞ | DEBUG nova.api.openstack.wsgi \[.* service nova\] Returning .* to user: .* |
| #5 | 956 | ∞ | DEBUG nova.compute.utils \[.* .* .* .*\] \[instance: .*\] Build of instance .* aborted: VolumeLimitExceeded: Maximum number of volumes allowed \(.*\) exceeded for quota 'volumes'. .* |
| #6 | 225 | 4.5029268227 | DEBUG nova.compute.multi_cell_list \[None .* demo admin\] Listed batch of .* results from cell out of .* limit. Returned .* total so far. {{\(pid=.*\) do_query .*}} |
| #7 | 191 | 3.0019511951 | DEBUG neutron.wsgi \[None .* demo admin\] .* returned with HTTP .* {{\(pid=.*\) __call__ .*}} |
| #8 | 62 | 2.7517881789 | DEBUG cinder.api.openstack.wsgi \[None .* demo admin\] Calling method 'all' {{\(pid=.*\) _process_stack .*}} |
| #9 | 149 | 2.2514634463 | DEBUG keystone.server.flask.request_processing.middleware.auth_context \[None .* service placement\] Validating token access rules against request {{\(pid=.*\) validate_allowed_request .*}} |
| #10 | 150 | 2.2514634463 | DEBUG keystone.server.flask.request_processing.middleware.auth_context \[None .* service placement\] Authenticating user token {{\(pid=.*\) process_request .*}} |

**(c) Term Frequency Analysis**

| | Term Frequency Analysis | | |
|---|---|---|---|
| Rank | Tmpl # | Frequency | Log Template Message |
| #19 | 225 | 1.625 | DEBUG nova.compute.multi_cell_list \[None .* demo admin\] Listed batch of .* results from cell out of .* limit. Returned .* total so far. {{\(pid=.*\) do_query .*}} |
| | | | ['listed', 'batch', 'results', 'cell', 'limit', 'returned', 'total', 'far'] |
| #20 | 216 | 1.66666666667 | DEBUG cinder.api.contrib.volume_transfer \[None .* demo admin\] Listing volume transfers {{\(pid=.*\) _get_transfers .*}} |
| | | | ['listing', 'volume', 'transfers'] |
| #21 | 207 | 1.75 | INFO cinder.volume.api \[None .* demo admin\] Get all volumes completed successfully. |
| | | | ['get', 'volumes', 'completed', 'successfully'] |
| #22 | 610 | 1.75 | DEBUG neutron.agent.linux.dhcp \[\-\] Done building host file .* {{\(pid=.*\) _output_hosts_file .*}} |
| | | | ['done', 'building', 'host', 'file'] |
| #23 | 956 | 1.77777777778 | DEBUG nova.compute.utils \[.* .* .* .*\] \[instance: .*\] Build of instance .* aborted: VolumeLimitExceeded: Maximum number of volumes allowed \(.*\) exceeded for quota 'volumes'. .* |
| | | | ['build', 'instance', 'maximum', 'number', 'volumes', 'allowed', 'exceeded', 'quota', ''] |
| #24 | 4 | 2.0 | DEBUG oslo_service.periodic_task \[None .* None None\] Running periodic task .* {{\(pid=.*\) run_periodic_tasks .*}} |
| | | | ['running', 'periodic', 'task'] |
| #25 | 38 | 2.0 | DEBUG nova.compute.resource_tracker \[None .* None None\] Total usable vcpus: .* total allocated vcpus: .* {{\(pid=.*\) _report_final_resource_view .*}} |
| | | | ['total', 'usable', 'total', 'allocated'] |

**(d) Final Rank Score**

| | Final Ranking | | |
|---|---|---|---|
| Rank | Tmpl # | Score | Log Template Message |
| #1 | 126 | 24 | DEBUG nova.compute.manager \[None .* None None\] Cleaning up deleted instances with incomplete migration {{\(pid=.*\) _cleanup_incomplete_migrations .*}} |
| #2 | 948 | 59 | DEBUG nova.api.openstack.wsgi \[.* service nova\] Returning .* to user: .* |
| #3 | 49 | 304 | DEBUG nova.compute.manager \[None .* None None\] CONF.reclaim_instance_interval <= .* skipping... {{\(pid=.*\) _reclaim_queued_deletes .*}} |
| #4 | 228 | 864 | DEBUG nova.compute.api \[None .* demo admin\] Searching by: {'deleted': False, u'project_id': .* {{\(pid=.*\) get_all .*}} |
| #5 | 956 | 1265 | DEBUG nova.compute.utils \[.* .* .* .*\] \[instance: .*\] Build of instance .* aborted: VolumeLimitExceeded: Maximum number of volumes allowed \(.*\) exceeded for quota 'volumes'. .* |
| #6 | 432 | 1501 | DEBUG neutron.agent.linux.dhcp \[\-\] Setting .* gateway for dhcp netns on net .* to .* {{\(pid=.*\) _set_default_route_ip_version .*}} |
| #7 | 21 | 2016 | DEBUG neutron.db.agents_db \[None .* None None\] Agent healthcheck: found .* active agents {{\(pid=.*\) agent_health_check .*}} |
| #8 | 946 | 2046 | DEBUG nova.api.openstack.compute.server_external_events \[.* service nova\] Unable to find a host for instance .* Dropping event .* {{\(pid=.*\) create .*}} |
| #9 | 79 | 2080 | DEBUG neutron.wsgi \[\-\] .* accepted .* {{\(pid=.*\) server .*}} |
| #10 | 947 | 2124 | INFO nova.api.openstack.wsgi \[.* service nova\] HTTP exception thrown: .* |

**Figure 6.** Ranking scores of the resource (block storage volume) limit exceed case.

## 3.4. Result Summary of All Scenarios

We present in Table 2 the evaluation results of all nine cases. All of the cases we tried contained key information in the log level of DEBUG or INFO implying that searching for ERROR logs will likely be of little help in the problem diagnosis. It is very difficult to come up with effective search terms without knowing the nature of the problem. On average, the log we are looking for are consistently ranked within top 10 out of more than 1000 log templates with the average rank of 3.8. This shows that the administrator can save significant amount of time during the problem diagnosis by looking at the top 10 log templates of the list recommended by Priolog.

**Table 2.** Summary of evaluation results.

| Case | Log Level | Log Message | Rank | Elapsed Time (s) | Log Lines |
|---|---|---|---|---|---|
| Component failure (`nova-compute`) | DEBUG | Seems service `nova-compute` on host * is down. | 2 | 1712 s | 34,903 |
| | | | 1 | 739 s | 5034 |
| Component failure (`neutron-dhcp-agent`) | DEBUG | No DHCP agents available, skipping rescheduling. | 2 | 1331 s | 5303 |
| | | | 8 | 900 s | 2987 |
| Launching very large (oversized) VM instance | INFO | Got no allocation candidates from the Placement API. This could be due to insufficient resources or a temporary occurrence as compute nodes start up. | 3 | 435 s | 1437 |
| | | | 3 | 550 s | 3165 |
| | | | 4 | 716 s | 2977 |
| Resource (volume count) limit exceeded | DEBUG | Build of instance * aborted: VolumeLimitExceeded: Maximum number of volumes allowed * exceeded for quota 'volumes'. | 5 | 598 s | 1818 |
| | | | 7 | 622 s | 2051 |

Table 2 also presents the time (in seconds) it took to complete the analysis to generate the final ranked list. Elapsed times of nine cases range from 7 min to 28 min. It is roughly proportional to the size of input log data as shown in the 'Log Lines' column. What is not included in the time measurement is the time it took to generate the log template lists which is a one-time process. The time cost of log template discovery varies greatly by the techniques. This state-of-the-art technique can produce the first version of log templates within several minutes and additional manual editing time has to be spent to fix any errors. We spent about 30 min to generate, correct, and prepare the log templates in semi-manual way. Based on this, the expected time cost of problem diagnosis using *Priolog* is at the level of tens of minutes at most.

## 4. Related Work

There has been extensive research in the detection of anomalies or outliers in logs using both machine learning approaches and using relations across multivariate time-series data in several application domains [14,17,18,24–31]. In this section, we review a set of representative examples of outlier detection applied to log analysis, and highlight a key focus of the contributions of our paper in the context of these rich body of prior art.

In the field of log-based anomaly detection, there are two types of anomalies—performance anomaly and behavioral anomaly. Most of the previous work focus on the detection of performance anomalies. In 2006, Mirgorodskiy et al. [29] developed a performance anomaly detection technique that is based on the traces of function running time. Function traces of HPC applications, collected by light-weight agent, are converted into vectors. Then, $k$th nearest neighbor search is used for a given trace to decide whether it is normal or abnormal. Xu et al. [30] proposed a technique of automatically creating features and applying PCA to detect anomalies. In order to create features, they parse the source code to understand the log templates, state variables and any identifiers. Two feature vectors, state ratio vector and message count vector, are constructed and they are fed into the PCA-based outlier detection technique to find anomalous log groups. Lou et al. [14] presents a technique that mines linear invariants from the logs. They first parse the logs into static log messages part and variable parts. Then, they group logs by the program variable values to form log groups. Per each group, they build message type count vector. From the invariant space of the matrix, they derive the execution flow invariants, and any violation of these invariants are considered as anomalies. ELT [25] proposes two stage approach for log-based troubleshooting. In the first stage, it uses hierarchical clustering on the message appearance vector to quickly group logs into anomalous and normal ones. Then, in the second stage, it uses message flow graph to further identify anomalous logs within the large normal cluster. Additionally, ELT supports the functionality of key message extraction by building difflogs which represents the set of log messages that do not appear in the normal logs. ELT can also do an invariant check of user-provided invariant rules. In lprof [27], authors have constructed the causal sequence of logs to build the model of correct execution sequence. Nandi et al. [17] have also tried to build the causal relationship of logs using a few heuristics to use them as a standard in detecting the deviation of log patterns. Logan [6] aims to help the admins to promptly perform the problem diagnosis and root cause analysis by performing automatic comparison of normal logs and problematic logs. Also, they narrow down the logs into the most likely meaningful regions of log.

Sabato [24] developed a method for ranking log messages by their importance to the users. It determines that the log is important if it appears more than expected. The objective of this work is closely in line with ours. However, we have learned that the log frequency method alone was insufficient to find the important logs. Our evaluation results shows that using the frequency of logs alone do not give high ranking of true positive logs. SALSA [32] parses logs using known keywords to extract states and constructs control-flow view and data-flow view of the Hadoop execution. These views are presented to the user for better understanding and problem diagnosis. Along with the state information it also extracts the duration of each state. They demonstrate that state duration information can be made into histogram by states per host and this histogram comparison allows them to identify

problematic hosts. GAUL [28] is for problem diagnosis using logs in storage systems. It uses logs to detect recurring problems and solutions.

In Deeplog [18], a deep learning approach based on LSTMs is presented for log analytics that is able to create workflows from logs, and give conditional probabilities of subsequent logs given current log based on an implicit finite state machine. Log sequences are treated in a manner similar to sequences of natural language sentences for deep learning purposes. Anomalies are detected, by detecting changes in the workflow expected sequence of logs and deviation from expected conditional probabilities. In addition, the method is able to perform online learning as combinations of logs change. This approach provides an improve way to detect anomalies in log sequences and relate these found anomalies them to expected workflows or sequences thereby providing additional insight to the developer. The method does not leverage domain knowledge or filter for false positives as done in our current work. The most recent work in the log anomaly detection field is the LogRobust [31]. Zhang et al. have developed a Bi-LSTM classification model from the fixed dimension semantic vectors of logs and improved the anomaly detection capability.

The goal of our work differs from most of the anomaly detection work in that we focus on developing set of techniques that help in the root cause identification rather than the anomaly detection. We assume that the fault has already happened and been detected by the administrator. Thus, Priolog is complementary to the anomaly detection work.

There are several research work focusing on the problem diagnosis based on the correlation analysis of metrics data, but not necessarily the logs. Priolog performs time series correlation analysis after converting the log streams to time series. CloudPD [33] is a cloud problem management framework that collects various metrics from the host server and VMs. Then, it uses a light-weight method such as kNN to detect potential anomalies. Once potential anomalies are detected, it undergoes correlation analysis between all metrics within a problematic host and between the same type of metrics across other hosts. If it is beyond some threshold, it considers as a problem. Once a problem is found, it matches the correlation signature to the known root-cause database which is built by experts. The similarity to our work is that it also uses correlations, but on system monitoring metrics. Jiang et al. [34] proposed an algorithm using the autoregressive models with exogeneous inputs (ARX) to discover hidden invariants between various system measurements. The example of invariants could be: the number of output requests of a load balancer must be equal to the number of input requests, the ratio of input request and the number of SQL query to the database is 2. Since it is difficult to holistically understand the distributed application, they say that it becomes easier if we monitor many of such small invariants, which hopefully characterizes the system well enough. PeerWatch [35] also uses correlation techniques to discover problems. It first collects various metrics such as CPU utilization, memory utilization, context switch, etc., from VMs running the same kind of applications. Then, it uses canonical correlation analysis (CCA) to discover highly correlated metrics between all pairs of VMs. CCA gives the list of metrics pairs in the order of correlation strength. PeerWatch first uses this information to identify which VM is faulty. Then, within a faulty VM, it identifies which attributes have changed significantly in terms of distribution.

## 5. Conclusions

We have presented a novel method, called Priolog, that is designed to quickly mine the most informative logs from the volume of log files for speedy problem diagnosis. The technique is based on the combination of log templates' temporal analysis, template frequency analysis, and word-level term frequency analysis. These three techniques, combined together, complement each other to generate accurate ranked list of most important logs. We have tested Priolog to the popular OpenStack platform under various failure scenarios. The results indicated that the Priolog's approach is promising for log-based problem diagnosis.

As a future direction, we plan to apply state-of-the-art techniques from the NLP domain such as the word embedding or n-gram analysis to improve the capability of Priolog in finding the most

important logs. For example, the word embedding allows us to search for the words having similar meaning to critical words such as 'failure' or 'errors'. Since word usage depends on the programmers who generated the log messages, word embedding allows us to learn the intended meaning of important terms. This would help us prioritize logs better. Another technique, n-gram analysis, can be used to learn the probabilistic model of word orders so that appearances of any abnormal word can be quantified and prioritized by the degree of statistical abnormality.

**Author Contributions:** Conceptualization, B.T. and P.K.; methodology, B.T.; software, S.P.; validation, B.T., S.P., and P.K.; resources, B.T.; data curation, S.P.; writing—original draft preparation, B.T.; writing—review and editing, B.T. and P.K.; visualization, S.P.; supervision, B.T.; project administration, B.T.; funding acquisition, B.T.

**Funding:** This research was funded by the BK21 Plus project (SW Human Resource Development Program for Supporting Smart Life) funded by the Ministry of Education, School of Computer Science and Engineering, Kyungpook National University, Korea (21A20131600005), the Basic Science Research Program through the NRF of Korea funded by the Ministry of Education (NRF-2017R1D1A3B03035777) and NRF funded by the Korean government(MSIT) (NRF-2019R1C1C1006990).

**Conflicts of Interest:** The authors declare no conflict of interest.

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
