# Peer review of "Priolog: Mining Important Logs via Temporal Analysis and Prioritization"

_sustainability, doi:10.3390/su11226306_

Round 1

Reviewer 1 Report

This paper uses three complementary techniques for identifying anomalies/outliers in log files: low correlation in time series, anomalous frequencies in log patterns, and infrequent appearance of terms within a log template. Each of these techniques covers a different background reason for the rise of abnormal entries in a typical log file for a computing system or application, so the authors use a compound index ---the product of the ranks--- to prioritize the potential alerts, giving equal weigh to each of the three independent analysis techniques.

The proposal is reasonably novel and interesting, the contents in the paper are well organised an presented, and the topic falls into an active research field. Though my assessment of this work is mostly positive, I still have a couple of concerns that must be addressed before the paper can be accepted:

1) One weakness in this work is that only two test cases are reported, conducted over logs collected from an operating Openstack environment. This is fair as a proof of concept, but does not give strong support to the effectiveness of the proposed algorithm. I would suggest the authors to run more experiments and, more importantly, to attempt to evaluate their method on a sounder basis: can false positives be avoided? what is the rate of false positives when only normal logs are used as input? Using only two test cases is not enough to demonstrate statistical correctness.

2) It is claimed that the method 'quickly mines the most informative logs ... for speedy problem diagnosis'. But nowhere in the paper one can find measurements or information about the speed of this technique in a specific case. Could the authors provide more detailed information about the running time/computational effort necessary when applying their method? It would greatly improve the paper quality, in my opinion.

Finally, the paper is nicely written and only a few typographical errors remain in the text. Please, revise these. 

Reviewer 2 Report

The paper represents a work on a current topic, in general, it is well written and easy to read.

I have some small concerns before being available as an article to be published.

Although the authors mention in the title and in the abstract the concept of log, it gives me the feeling that they speak of the logs in a slightly vague way, it would be very interesting to indicate more clearly the objective of this work as well as the contribution to the research community.

Introduction
References 13-16: old techniques, comparisons should be carried out against more current references.
12, this one seems interesting and since it is deep learning and more current, the explanation should be extended.
define the first abbreviation in VM

page 5, this type of quote seems strange without indicating anything else about it.

section 4.3: I can't believe that all the references are as old as the ones indicated in this section.

page 13: "... from NLP domain to improve the capability ..."
could it be extended? and maybe applied to other areas?

Reviewer 3 Report

The authors should carefully read the scope and aims of this journal: https://www.mdpi.com/journal/sustainability/about

This manuscript is related to informatics and data analysis.

Author Response

Thank you for pointing out the topic area of this journal.

However, I would like to bring to your attention that we submitted the paper to one of the special issue in the sustainability journal, named "Advanced IT based Future Sustainable Computing".

https://www.mdpi.com/journal/sustainability/special_issues/it_based_future_sustainable_computing

Among the topics of interests, we are roughly aiming at combinations of these topics.

Optimization, machine learning, prediction and control, decision support systems for FSC Management in memory, disk, storage and other peripheral devices with cloud computing Monitoring and visualization methodologies and tools in FSC Security and safety for FSC

We hope this addresses your concern of relevancy. Thank you.

Round 2

Reviewer 1 Report

In this revision the authors have properly addressed my previous concerns, mostly related to the lack of testing. Now, the paper has been improved and contains stronger support for the presented technique. I would like to thank the authors for the effort. 

Reviewer 3 Report

The authors have adequately justified the relevance of their paper and the examined research topic to the scope of the journal. The reviewer does not have major concerns regarding the proposed analysis and performance evaluation.